# Research on optimization of transportation routes for infectious medical waste

**Chen Hao**[1], **LI Libo**[2], **Qin Xuerui**[1]*, **Wang Wenxian**[3]

1 School of Economics and Business Administration, Yibin University, Yibin, Sichuan, China,
2 Department of Tourism Management, The Graduate School of Pai Chai University, Daejeon, Korea,
3 School of Rail Transportation, Wuyi University, Jiangmen, Guangdong, China

* 2019114013@yibinu.edu.cn

## Abstract

During the pandemic, the amount of infectious medical waste has increased dramatically. Currently, the medical waste recycling process generally suffers from defects such as long distances, high costs, and a lack of emergency response mechanisms. This paper addresses the problem of medical waste collection and route optimization for regions with multiple vehicle types and stages. It comprehensively considers factors such as transportation costs, distance, vehicle allocation, and contamination risks during the collection and distribution of medical waste. The goal is to minimize transportation costs and risks, with constraints including uniqueness, connectivity between nodes, and vehicle load capacity. A segmented collection approach is used to model the medical waste collection process. An optimization method for medical waste collection site selection and vehicle routing is proposed. Given the NP-hard nature of the problem, a location allocation method based on minimum envelope clustering analysis is employed, and an improved NSGA-II algorithm incorporating a fast non-dominated sorting mechanism is designed to obtain Pareto optimal solutions. Comparing with the results of traditional genetic algorithms through simulation, the results show that using the improved NSGA-II to solve practical problems: 1. When the production of medical waste is flat (1 disposal center, 4 backup transfer points, 58 producing points), the total cost is reduced by 13.94%, the total mileage is reduced by 7.17%, the full load rate is increased by 6.14%, and the convergence time is 26 seconds. 2. When the production of medical waste increased significantly (1 disposal center, multiple backup transfer points, 58 producing points), the total cost, total mileage, and transportation risk were reduced by 9.50%, 10.35%, and 2.03%, respectively, and the full load rate increased by 5.98%. The final results also indicate that compared to the results obtained by traditional genetic algorithms, the improved NSGA-II algorithm performs better in solving the optimization problem of infectious medical waste transportation routes.

**Data availability statement:** All relevant data are within the manuscript and its Supporting Information files.

**Funding:** The authors would like to express gratitude for the support from Yibin University (No.412-2022QH23 to Hao Chen), Sichuan Provincial Key Laboratory of Automobile Measurement and Control and Safety (No.QCCK2019-006 to Hao Chen), Yibin Federation of Social Science Associations (No. 2024YBSKL81 to Hao Chen), Collaborative Innovation Center Project for Shipping and Logistics in the Upper Reaches of the Yangtze River (No. XTCX2024B11 to Wenxian Wang), and Jiangmen Science and Technology Planning Project (No.2021030101730004367 & No.2022030100950003858 to Wenxian Wang).

**Competing interests:** The authors have declared that no competing interests exist.

## 1 Introduction

The management and recycling of medical waste has gradually become one of the important environmental, health, and social issues. Due to population growth and urbanization, especially the increase in the number of COVID-19 and other epidemics, the amount of medical waste is also continuously increasing (as shown in Fig 1), and the disposal of medical waste has become a focus [1].

Medical waste refers to the waste produced by medical and health institutions in medical, preventive, healthcare, and other related activities, which has direct or indirect infectivity, toxicity, and other harmful characteristics, such as space pollution, acute transmission, and potential transmission [2]. With the progress of society and the improvement of medical standards, especially when encountering global infectious diseases such as COVID-19, the number of people seeking medical treatment will increase, leading to an increase in the amount of medical waste. How to efficiently and reasonably dispose of the medical waste is a problem that countries around the world are facing and must solve. Once medical waste is improperly disposed of, it may pose a threat to individuals who come into direct or indirect contact, increasing the risk of disease transmission [3]. Therefore, improving the efficiency and safety of medical waste recycling management plays an important role in social development.

The design of a medical waste recycling system refers to the process where, within a certain research scope, all nodes where medical waste is generated are collected by professional recycling vehicles to disposal points under the constraint of vehicle load capacity. Therefore, the key to designing a medical waste recycling system lies in establishing an appropriate number and location of medical waste transfer centers, and then determining the transportation routes for all vehicles involved in recycling, so that the overall cost of the waste recycling plan is minimized. At the same time, the operational routes of the recycling vehicles should minimize the risk of infection spread to surrounding residents. It can be seen that the core issue of the medical waste recycling system design addressed in this paper is the integrated optimization of the selection of recycling centers and vehicle routing. An important premise for the successful implementation of this method is that basic data such as the locations of medical waste generation nodes, the number of delivery vehicles, their load capacity, the infection range, and the number of residents within the infection zone are given values. The main contribution of this paper is reflected in:

(1) During the period when a large amount of medical waste is produced, the recycling process involves multiple vehicles, high costs, and high infectious risks. Therefore, this paper considers the selection of medical waste transfer points and transportation route optimization under the risk of infection, making the recycling plan more in line with the actual situation.

(2) This paper establishes a multi-objective 0–1 decision integer programming model for the selection of medical waste transfer points and the optimization of transportation routes, with transportation costs and infection risks as optimization objectives.

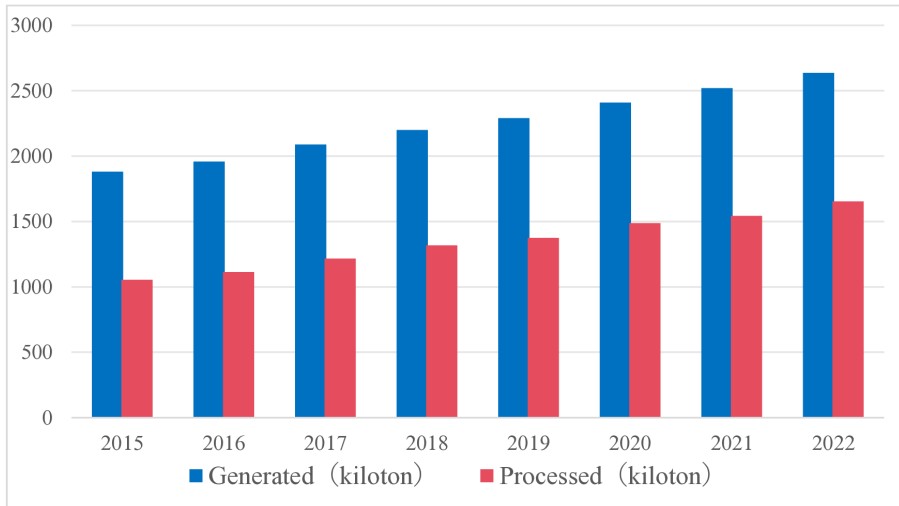

**Fig 1. The amount of medical waste generated and processed in China.**

(3) This paper proposes an improved NSGA-II, which proves its strong applicability in model solving by taking Yibin City during the COVID-19 as an example, and has certain advantages compared to traditional genetic algorithms.

The rest of this paper is arranged as follows: Section 2 reviews relevant literature. Section 3 formally describes this optimization problem in mathematical language. Section 4 constructed an optimization model. In Section 5, the envelope clustering algorithm and improved NSGA-II were designed. Section 6 conducted an example analysis to optimize the recovery plan for infectious medical waste in Yibin City during the COVID-19 period and obtained experimental results. Finally, the conclusion is presented in Section 7.

## 2 Literature review

Medical waste belongs to hazardous waste. In recent years, scholars from various countries have conducted extensive research in the fields of risk measurement and location route optimization of medical waste transportation. Relatively speaking, research on reverse logistics of medical waste in foreign countries is much more mature than in China, and related laws, regulations, and disposal processes are also relatively complete. Ozbek [4] took the waste produced in the department of stomatology as the research object, analyzed it in Türkiye, and carefully summarized the particularity of the recycling and disposal process of medical waste in the department of stomatology. Stancu Malaysia [5] constructed a reverse logistics network planning model with cost as the sole objective in an uncertain environment. Stern [6] established a model with the two objectives of minimizing fixed costs and maximizing service capacity for the construction of service stations in reverse logistics networks, mainly studying the optimization problem of station networks. Listers [7] considered the uncertainty of facility construction costs and established a stochastic mixed integer programming model based on the analysis of reverse logistics networks. Aylin [8] took Istanbul city as the research object and comprehensively considered the factors of recycling points, disposal points, and transportation vehicles in the reverse logistics recycling system, optimizing the recycling and transportation routes of medical waste. Qin [9] proposed a fuzzy chance constrained model based on reverse logistics networks and solved the model using intelligent algorithms. Based on the theory of system dynamics, He [10] selected indicators related to reverse logistics systems for evaluation among the three systems of society, economy, and logistics, and established an evolution model of the reverse logistics network for medical waste. Nie [11] took rural areas as the research object, considered issues such as recycling volume and distance, and selected

storage locations for medical waste. She established a dual objective model with the lowest cost and maximum recycling capacity, and solved the model using genetic algorithm to verify the feasibility of the model and algorithm.

In the study of transportation routes for waste, Dong [12] constructed a multi-objective mixed integer programming model with the goal of minimizing total economic cost and total risk, and obtained the optimal medical waste management system network after grouping. Shi [13] took Dalian City as the research object, combined with the number of logistics nodes in the road network, established a set coverage model, and obtained the minimum number of transfer points and the shortest recycling route in the area. He [14] constructed a multi-layer reverse logistics network and used greedy algorithms to solve a dual objective integer programming model. He planned the capacity and location of producing points, transfer points, and disposal centers in the network, and obtained the minimum cost scheme. Osaba [15] designed a multi-objective and multi constraint vehicle routing model and established an improved discrete bat algorithm, providing a new solution for the medical waste collection problem in Bizcaya, Spain. Le [16] proposed a generalized vehicle routing model that includes multiple transfer points, aggregation points, and non-uniform vehicles within a time window. Through the model, the optimal path for solid waste recycling vehicles in the streets of Danang, Vietnam was obtained, effectively reducing the total driving distance of vehicles. Liu [17] used the periodic recycling strategy of medical waste as the research background and solved the medical waste recycling route model using domain search algorithms. Ma [18] considered the urban traffic saturation and environmental costs during the collection, disposal, and transportation of solid waste, established a bi level programming model, and determined the location of the recycling station and transportation route with the lowest total cost in the urban solid waste disposal system. Alshraideh [19] designed a stochastic model for medical waste collection in northern Jordan and proposed a route scheduling model that minimizes total travel distance, ultimately minimizing transportation costs and carbon emissions. Wang [20] defined the infection process and the transmission risk of infected populations, set up multiple disposal centers, developed a dual objective route model for medical waste collection to optimize infection risk and transportation costs, and obtained Pareto solution by combining weighted economic constraint method.

Table 1 provides a detailed comparison between current research and previous studies on optimizing vehicle delivery routes. From the existing research results, scholars at home and abroad have conducted a lot of research on waste site selection and path optimization, and there is also sufficient research on the site selection and path optimization of

**Table 1. Comparison of research contributions with existing studies.**

| Authors | Cost | Capacity | Risk | Mileage | Objective | Method | Environment |
|---|---|---|---|---|---|---|---|
| Stancu (1976) | √ | √ | | | Multi | MILP | Deterministic |
| Stern (1981) | | | √ | | Single | MILP | Deterministic |
| Listers (2005) | √ | | | | Single | MILP | Stochastic |
| Aylin (2008) | √ | | | | Single | MILP | Stochastic |
| Nie (2018) | √ | √ | | | Multi | GA | Deterministic |
| Dong (2022) | √ | | √ | | Multi | GA | Deterministic |
| Shi (2011) | | | | √ | Single | ACS | Deterministic |
| He (2007) | √ | | √ | | Multi | Greedy | Deterministic |
| Osaba (2018) | √ | | | √ | Multi | Bat | Deterministic |
| Le (2016) | √ | | | √ | Multi | MILP | Deterministic |
| Liu (2016) | | | | √ | Single | Neighbor | Deterministic |
| Ma (2016) | √ | | | √ | Multi | MILP | Deterministic |
| Alsh (2017) | √ | | | √ | Multi | MILP | Stochastic |
| Wang (2023) | √ | | √ | | Multi | GA | Deterministic |
| This article | √ | √ | √ | √ | Multi | NSGA-II | Deterministic |

hazardous waste disposal centers. However, although medical waste is also classified as hazardous waste, there are still differences in essence. On the one hand, hospitals need to be considered in the disposal process of medical waste, and on the other hand, the disposal center for infectious waste requires a dedicated disposal center, which is different from other waste disposal centers. Therefore, the research on site selection and path optimization of medical waste recycling centers is different from other waste disposal methods in practical situations. At present, there is a lack of quantitative research literature on the site selection and path optimization of medical waste disposal, and there are more qualitative studies. Especially, few scholars have paid attention to the more detailed field of infectious medical waste disposal.

In summary, the existing research results have the following shortcomings: Firstly, a risk measurement model for the transmission of infectious medical waste. Not taking into account the environmental transmission characteristics of the virus and the structural characteristics of the risk source; Secondly, the location route optimization model ignores the impact of increasing transportation mileage and volume on the optimization objectives; Finally, when considering multi-objective combination optimization problems such as transportation costs and risks, the weight ratio of optimization objectives lacks basis, and stability and sensitivity analysis are lacking in solving.

Therefore, based on the environmental transmission characteristics of viruses, this paper proposes a multi-objective, three-dimensional risk measurement model that considers the risk of virus transmission and the generalized system cost. Moreover, this paper introduces the impact of road traffic volume on the risk of virus transmission, and constructs a model of transfer point locations – transportation routes with the goal of minimizing total system cost and transmission risk. Finally, Pareto solution is obtained based on minimum envelope clustering and NSGA-II. Taking the optimization of infectious medical waste recycling plan during COVID-19 in Yibin City as an example, this paper verifies the effectiveness of the model and algorithm.

## 3 Problem description

### 3.1 Recycling system of medical waste

Infectious medical waste disposal centers are usually built in remote areas with fewer populations. Considering convenience and economy, multiple transfer points are generally established within urban areas as short-term distribution centers. Collect waste using small capacity vehicles first, store it at the transfer point, and then transport it to the processing point using large capacity vehicles.

Medical waste refers to the waste generated by medical and health institutions in medical, preventive, healthcare, and other related activities, which has direct or indirect infectivity, toxicity, and other hazards. The recycling of such medical waste generally has two stages. The first stage is from medical institutions to transfer points, and the second stage is from transfer points to medical waste disposal centers. Considering the transportation risks of medical waste, it is necessary to equip rescue centers in certain areas to respond to sudden waste leakage situations. The specific recycling system is shown in Fig 2.

From Fig 2, it can be seen that the infectious medical waste transportation system is architecture in above the urban road network containing emergency system, including the produce points (medical institutions), transfer points, and disposal centers of medical waste. In the first stage, transportation vehicles depart from the transfer point, collect medical waste from several production points, and then return to the transfer point for temporary storage; The second stage is similar to the first stage, starting from the disposal center, passing through several transfer centers, and returning to the disposal center. If a leakage accident occurs during transportation, the rescue center in the area will quickly handle it to control the spread of the virus. Therefore, the problem of transfer points location – transportation route decision is a two-stage multi-objective optimization problem, that involves collaborative optimization of multiple links such as rescue centers, medical institutions, transfer points, and disposal centers in the event of a virus leak.

In summary, the design of infectious medical waste transportation systems faces four difficulties. Firstly, due to the suddenness and instability of the epidemic, it is difficult to determine the maximum load of the transportation system;

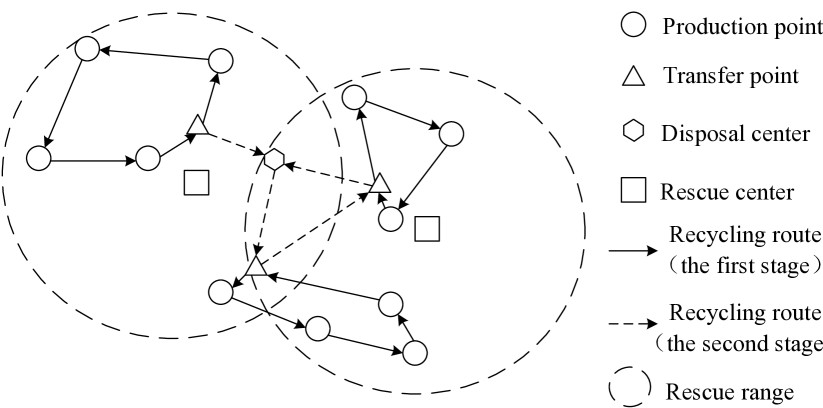

**Fig 2. Infectious medical wastes transport system.**

Secondly, it is difficult to determine the optimal construction location and service target of the transfer point; Thirdly, encountering a large-scale epidemic, such as COVID-19, can lead to a sharp increase in infectious medical waste, and there are strict requirements for the disposal facilities and sites of such medical waste, which limits the disposal capacity of a city; Fourthly, when considering the problem of minimizing the risk of infection and optimizing the location path, it is difficult to quantify the weight relationship between multiple objectives, making it difficult to determine the quality of the final transportation plan during the solving process.

### 3.2 Risk measurement of medical waste

Urban garbage removal refers to the entire process of transporting garbage from the generation point to the garbage disposal site, which includes three stages: collection, transportation, and treatment. Infectious risk refers to the possibility of virus leakage during transportation of infectious medical waste, which can be transmitted to surrounding residents. This paper refers to the measurement method proposed by LI Xue [17] and defines the transportation risk of infectious medical waste as: The product of the resident number exposed to the virus transmission area along the route and the virus concentration in the environment after a leakage accident occurs during transportation.

It can be defined that vehicles transporting infectious medical waste are risk points, and the trajectory of the risk point is a line segment. Considering the characteristics of infectious viruses, their diffusion process can be described as follows: starting from vehicles, the virus spreads in a three-dimensional manner with a certain diffusion radius. Within a certain period of time, the virus will roughly form a region similar to a half cylinder, as shown in Fig 3. It can be found that the risk of virus transmission mainly depends on parameters such as spread radius $R$, transportation path length $L_{ij}$, and vehicle height $H$.

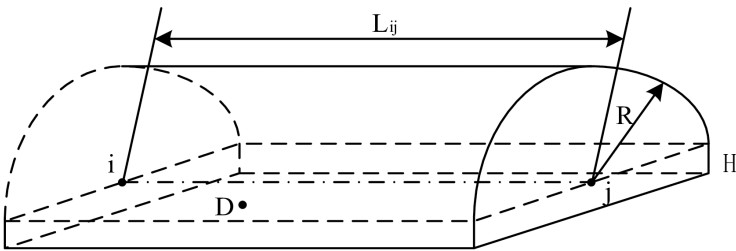

**Fig 3. Schematic diagram of virus spread range.**

# 4 Model formulation

## 4.1 Symbol

The symbols used in this paper are explained in Table 2.

## 4.2 Assumptions

The location-path optimization problem for the recycling of infectious medical waste can be described as: designing a reasonable number of transfer points, starting from several transfer points by one or more medical special vehicles of the same model, passing through several hospital institutions in sequence, to transport all medical waste to the transfer point. Then, organizing transportation vehicles with larger carrying capacity at the transfer point to transport the medical waste from the transfer point to the nearest disposal center. This paper makes the following assumptions in order to simplify the solution model while being in line with the actual situation:

(1) There is one medical waste disposal center, multiple transfer points, and multiple producing points in the area, and the disposal center is equipped with different types of vehicles to meet the transportation needs;

(2) Each producing point can only be served by one transport vehicle, and the amount of medical waste generated is less than the load capacity of the transport vehicle;

(3) Each medical institution can only be served by the same vehicle, and vehicles are not allowed to be overloaded;

(4) The distance between nodes is calculated based on Euclidean distance, and there is no limit to the capacity on the route;

(5) Not considering the impact of external environments such as traffic congestion;

(6) When medical waste leaks, the virus spreads evenly.

## 4.3 Constructing the objective function

The optimization objective of the model is: during the transportation process, vehicles complete the collection of waste from all medical institutions, temporarily store it at the transfer point, and then deliver it to the disposal center. This process has the lowest cost and minimal transportation risk.

**4.3.1 Objective function 1: Lowest cost.** The total cost of the transportation system mainly includes the construction cost of backup transfer points, transportation cost, and vehicle usage cost, with other costs accounting for a relatively low proportion that can be ignored. Therefore, the objective function with the lowest cost can be expressed as:

$$f_1 = \min\left(C_1 + C_2 + C_3\right) \tag{1}$$

(1) Fixed construction cost of backup transfer center

$$C_1 = \sum_{g \in G} \alpha_g \cdot FC_g \tag{2}$$

(2) Transportation costs

$$C_2 = \sum_{i,j \in V} \beta_{ij} \cdot L_{ij} \cdot Q_{ij} + \sum_{i \in V} \sum_{g \in G} \alpha_g \beta_{ig} \cdot L_{ig} \cdot Q_{ig} + \sum_{g \in G} \sum_{s \in S} \alpha_g \beta_{gs} \cdot L_{gs} \cdot Q_{gs} \tag{3}$$

**Table 2. Symbols used in the paper.**

| Symbols | Parameter |
|---|---|
| $N = V \cup G \cup S$ | Node set |
| $V = \{1, \cdots, i, \cdots, j, \cdots\}$ | Node set of producing points |
| $G = \{1, \cdots, g, \cdots, g', \cdots\}$ | Node set of backup transfer points |
| $S = \{1, \cdots, s, \cdots, s', \cdots\}$ | Node set of disposal center |
| $K = \{1, \cdots, k, \cdots\}$ | Node set of vehicles |
| E | Set of point-to-point route |
| $D = \{1, \cdots, d\}$ | Location of rescue center(handling infectious incidents) |
| H | Vehicle height |
| R | Infection radius |
| $\mu$ | Diffusion velocity |
| T | Diffusion time |
| $d_{Dij}$ | The shortest distance from rescue center route$(i, j)$ |
| $v_D$ | Average speed of rescue vehicles |
| $Q_{ij}$ | The weight of medical waste infected on route $(i, j)$ |
| $F_D$ | The rescue capabilities that rescue center D can provide |
| $L_{ij}$ | Length of route $(i, j)$ |
| $O_{ij}$ | Spread range of the virus on route $(i, j)$ |
| $\theta$ | Probability of medical waste carrying infectious viruses |
| $P_{ij}$ | Population density on the route$(i, j)$ |
| M | Population within the scope of virus spread |
| $TR_{ij}$ | Transportation risks on the route$(i, j)$ |
| $FC_g$ | Construction cost of the transfer center g |
| $FC_k$ | Fixed transportation cost of vehicle k |
| $q_i$ | The weight of medical infections produced in medical institution i |
| $q_g$ | The weight of medical infectious materials stored in transfer center g |
| $Z_k$ | Maximum capacity of vehicle k |
| $Z_g$ | Maximum capacity of transfer center g |
| $Z_s$ | Maximum capacity of the disposal center s |
| C | Unit transportation cost |
| $R_D$ | Rescue radius of rescue center D |
| $\alpha_g$ | 0-1 variable, If a transfer point is established at node g,$\alpha_g = 1$, otherwise, $\alpha_g = 0$ |
| $\beta_{ij}$ | 0-1 variable, If the vehicle passes through route$(i, j)$ ,$\alpha_g = 1$, otherwise, $\alpha_g = 0$ |
| $\gamma_k$ | 0-1 variable, If vehicle k is used for transportation,$\alpha_g = 1$, otherwise, $\alpha_g = 0$ |

(3) Fixed usage cost of vehicles

$$C_3 = \sum_{k \in K} \gamma_k \cdot FC_k$$

(4)

#### 4.3.2 Objective function 2: Minimize transport risk.
The total transportation risk is represented by the product of transportation risk, diffusion time, and population density along each section of the route.

The radius of the diffusion range is:

$$R = \mu \cdot T$$

(5)

Among them, the diffusion time $T$ can be expressed as:

$$T = {}^{d_{Dij}}\!/_{v_D} + {}^{Q_{ij}}\!/_{F_D}$$

(6)

The cross-sectional area of the diffusion range is:

$$S = {}^1\!/_2 \pi R^2 + 2R \cdot H$$

(7)

$L_{ij}$ is the length of the route $(i, j)$, and the volume of diffusion range can be expressed as:

$$O_{ij} = S \cdot L_{ij} = \left( {}^1\!/_2 \pi R^2 + 2R \cdot H \right) \cdot L_{ij}$$

(8)

The concentration of the virus is represented by the population within the current diffusion range:

$$M = \pi \cdot \mu^2 \cdot T^2 \cdot \theta \cdot P_{ij} \cdot L_{ij}$$

(9)

Based on equations (4) and (5), the transportation risk on the route $(i, j)$ can be expressed as:

$$TR_{ij} = \frac{4\theta \cdot Q_{ij} \cdot P_{ij}}{\pi \cdot \mu^2 \cdot T^2 + 4\mu \cdot T \cdot H}$$

(10)

Transportation risk is represented by:

$$f_2 = \min \left( \sum_{i,j \in N} TR_{ij} \cdot \mu T \cdot P_{ij} \right) = \min \frac{\sum\limits_{i,j \in N} 4\theta \cdot Q_{ij} \cdot P_{ij} \cdot \mu T}{\pi \cdot \mu^2 \cdot T^2 + 4\mu \cdot T \cdot H}$$

(11)

#### 4.3.3 Rescue model in case of infectious medical waste leakage.
When medical waste belongs to highly contagious, widely contagious, and rapidly contagious infectious substances, m is the location of infectious waste leakage, $m \in E_{ij}$. At this point, add a new objective function $f_3$: the fastest rescue speed, i.e., the shortest rescue distance:

$$\min f_3 = \max \left\{ \frac{L_{Di} + L_{im} \cdot \beta_{ij}}{v_D}, \frac{L_{Dj} + L_{jm} \cdot \beta_{ij}}{v_D} \,\Big|\, Q_{ij} \neq 0 \right\}$$

(12)

## 4.4 Constraints

(1) Uniqueness constraint:

The uniqueness constraint of the model includes the following two aspects: Firstly, for any medical institution node, there is only one transport vehicle to transport it, as shown in formula (13); secondly,Vehicles departing from the transfer point to collect waste at medical institutions and ultimately returning to the corresponding transfer point, As shown in formulas (14) and (15).Vehicles that depart from the disposal center to load waste at the transfer point and ultimately return to the corresponding disposal center, As shown in formulas (16) and (17):

$$\sum_{i,j\in V}\sum_{k\in K}(\beta_{ij}\cdot\gamma_k)=1 \tag{13}$$

Vehicles departing from the transfer point to collect waste at medical institutions and ultimately returning to the corresponding transfer point:

$$\sum_{i\in V}\sum_{g\in G}\sum_{k\in K}(\alpha_g\cdot\beta_{gi}\cdot\gamma_k)=1 \tag{14}$$

$$\sum_{j\in V}\sum_{g\in G}\sum_{k\in K}(\alpha_g\cdot\beta_{jg}\cdot\gamma_k)=1 \tag{15}$$

Vehicles that depart from the disposal center to load waste at the transfer point and ultimately return to the corresponding disposal center:

$$\sum_{g\in G}\sum_{s\in S}\sum_{k\in K}(\alpha_g\cdot\beta_{sg}\cdot\gamma_k)=1 \tag{16}$$

$$\sum_{g'\in G}\sum_{s\in S}\sum_{k\in K}(\alpha_{g'}\cdot\beta_{g's}\cdot\gamma_k)=1 \tag{17}$$

(2) Logical constraints

Connectivity constraints between nodes. The inter node connectivity constraint refers to the requirement that any two nodes should have connectivity, as shown in formula (18). There is no mutually ineffective transportation between each transfer center and each processing center, as shown in formula (19). For any medical institution, there is only one vehicle that comes to serve, and at the same time, only this vehicle leaves the medical institution, as shown in formula (20).

$$\sum_{i,j,g,s\in N}\sum_{k\in K}\alpha_g\cdot\beta_{ij}\cdot\gamma_k-\sum_{i,j,g,s\in N}\sum_{k\in K}\alpha_g\cdot\beta_{ji}\cdot\gamma_k=0 \tag{18}$$

No invalid transportation shall be carried out between each transfer point and each processing center:

$$\sum_{g,g'\in N}\beta_{gg'}=\sum_{s,s'\in N}\beta_{ss'}=0 \tag{19}$$

For any medical institution with only one vehicle to collect waste and only one vehicle to leave the institution:

$$\sum_{i,j\in V}\sum_{g\in G}\sum_{k\in K}\beta_{gi}\cdot\gamma_k + \sum_{i,j\in V}\sum_{k\in K}\beta_{ji}\cdot\gamma_k = 1 \tag{20}$$

$$\sum_{i,j\in V}\sum_{k\in K}\beta_{ij}\cdot\gamma_k + \sum_{i,j\in V}\sum_{g\in G}\sum_{k\in K}\beta_{ig}\cdot\gamma_k = 1 \tag{21}$$

(3) Capacity constraints

Conservation of weight of medical waste on transportation routes:

$$\sum_{i,j,n\in N}\sum_{k\in K}Q_{ij}\cdot\gamma_k + q_j = \sum_{i,j,n\in N}\sum_{k\in K}Q_{jn}\cdot\gamma_k \tag{22}$$

The total weight of medical waste shall not exceed the maximum load capacity of the vehicle. At transfer points and disposal centers, the total weight of medical waste does not exceed the capacity limit:

$$\sum_{i,j\in V}\sum_{k\in K}\beta_{ij}\cdot\gamma_k\cdot q_j \leq Z_k \tag{23}$$

$$\sum_{g\in G}\alpha_g\beta_{jg}\left(\sum_{i,j\in V}\sum_{k\in K}\beta_{ij}\cdot\gamma_k\cdot q_i\right) \leq Z_g \tag{24}$$

$$\sum_{s\in S}\sum_{g\in G}\alpha_g\beta_{jg}\beta_{gs}\left(\sum_{i,j\in V}\sum_{k\in K}\beta_{ij}\cdot\gamma_k\cdot q_i\right) \leq Z_s \tag{25}$$

At the same time, it is necessary to add constraints that the amount of leaked infectious medical waste cannot exceed the processing capacity of the rescue center:

$$\sum_{i,j\in V}\sum_{k\in K}\beta_{ij}\cdot q_i\cdot\gamma_k \leq F_D \tag{26}$$

(4) Limitations on the scope and quantity of rescue centers:

All nodes and routes are within the rescue range of a certain rescue center:

$$d_{Dij} \leq R_D \tag{27}$$

The number of rescue centers is greater than or equal to 1:

$$\sum D \geq 1 \tag{28}$$

In summary, the recycling optimization model for the infectious medical waste is as follows:

$$f_1 = \min\left(C_1 + C_2 + C_3\right)$$

$$f_2 = \min \frac{\sum\limits_{i,j\in N} 4\theta \cdot Q_{ij} \cdot P_{ij} \cdot \mu T}{\pi \cdot \mu^2 \cdot T^2 + 4\mu \cdot T \cdot H}$$

$$f_3 = \min\left\{\max\left\{\frac{L_{Di} + L_{im}\cdot\beta_{ij}}{v_D}, \frac{L_{Dj} + L_{jm}\cdot\beta_{ij}}{v_D}\,\Big|\,Q_{ij}\neq 0\right\}\right\}$$

*s.t.*

$$\sum_{i,j\in V}\sum_{k\in K}(\beta_{ij}\cdot\gamma_k) = 1$$

$$\sum_{i\in V}\sum_{g\in G}\sum_{k\in K}(\alpha_g\cdot\beta_{gi}\cdot\gamma_k) = 1$$

$$\sum_{j\in V}\sum_{g\in G}\sum_{k\in K}(\alpha_g\cdot\beta_{jg}\cdot\gamma_k) = 1$$

$$\sum_{g\in G}\sum_{s\in S}\sum_{k\in K}(\alpha_g\cdot\beta_{sg}\cdot\gamma_k) = 1$$

$$\sum_{g'\in G}\sum_{s\in S}\sum_{k\in K}(\alpha_{g'}\cdot\beta_{g's}\cdot\gamma_k) = 1$$

$$\sum_{i,j,g,s\in N}\sum_{k\in K}\alpha_g\cdot\beta_{ij}\cdot\gamma_k - \sum_{i,j,g,s\in N}\sum_{k\in K}\alpha_g\cdot\beta_{ji}\cdot\gamma_k = 0$$

$$\sum_{g,g'\in N}\beta_{gg'} = \sum_{s,s'\in N}\beta_{ss'} = 0$$

$$\sum_{i,j\in V}\sum_{g\in G}\sum_{k\in K}\beta_{gi}\cdot\gamma_k + \sum_{i,j\in V}\sum_{k\in K}\beta_{ji}\cdot\gamma_k = 1$$

$$\sum_{i,j\in V}\sum_{k\in K}\beta_{ij}\cdot\gamma_k + \sum_{i,j\in V}\sum_{g\in G}\sum_{k\in K}\beta_{ig}\cdot\gamma_k = 1$$

$$\sum_{i,j,n\in N}\sum_{k\in K}Q_{ij}\cdot\gamma_k + q_j = \sum_{i,j,n\in N}\sum_{k\in K}Q_{jn}\cdot\gamma_k$$

$$\sum_{i,j\in V}\sum_{k\in K}\beta_{ij}\cdot\gamma_k\cdot q_j \leq Z_k$$

$$\sum_{g\in G}\alpha_g\beta_{jg}\left(\sum_{i,j\in V}\sum_{k\in K}\beta_{ij}\cdot\gamma_k\cdot q_i\right) \leq Z_g$$

$$\sum_{s\in S}\sum_{g\in G}\alpha_g\beta_{jg}\beta_{gs}\left(\sum_{i,j\in V}\sum_{k\in K}\beta_{ij}\cdot\gamma_k\cdot q_i\right) \leq Z_s$$

$$\sum_{i,j\in V}\sum_{k\in K}\beta_{ij}\cdot q_i\cdot\gamma_k \leq F_D$$

$$d_{Dij} \leq R_D$$

$$\sum D \geq 1$$

## 5 Solution algorithm

This paper models the location and routing optimization of infectious medical waste recycling, which leads to an explosive increase in computational complexity as the number of nodes grows, causing the calculations to be plagued by

dimensional issues. Additionally, the model incorporates time windows, damage rates, and time-varying transportation speeds, containing many variables and constraints, resulting in a large-scale, multi-objective optimization problem. To address this, a heuristic search algorithm suitable for solving this model is proposed.

The NSGA-II algorithm inherits the "survival of the fittest" group search mechanism from basic genetic algorithms, offering good parallel computing and global search capabilities. Furthermore, the algorithm introduces an elitism strategy to ensure that certain high-quality individuals in the population are not discarded during the evolution process. By using a crowding distance and a crowding comparison operator, it overcomes the limitation of traditional multi-objective algorithms, which require manually setting the objective weights. This ensures that the individuals in the approximate Pareto domain are evenly distributed across the entire Pareto domain, thereby maintaining the diversity of the population.

Based on this, considering the uncertainty of transfer center locations in the research problem, which can lead to a large number of nodes and excessively long solving time and computational load, a two-stage solving strategy is designed. The first stage involves a location-allocation algorithm based on minimum enclosing clustering analysis, while the second stage uses an improved NSGA-II algorithm for routing optimization. As a result, the problem in this paper is transformed into a 2E-MOLRP problem (two-echelon multi-objective localization path problem), which further reduces the complexity of solving the problem.

## 5.1 Location-distribution algorithm based on minimum envelope clustering analysis

At this echelon, it is necessary to determine the construction location of the transfer point and determine the specific service recipients (medical institutions) for each vehicle at the transfer point. The dissimilarity between two objective functions is defined as the distance function between two feature spaces. The $i$ th variable of $x^*$ and $y^*$ is defined as $x_i^*$ and $y_i^*$, and the set of variables that can change is $q^*$. The distance function can be expressed as [21]:

$$d(x^*, y^*) = \sqrt{\sum_{i=1}^{q^*} (x_i^* - y_i^*)^2}$$

(29)

**5.1.1 Euclidean distance with the property of value.** Taking the quantity and unit freight rate of waste, and fixed cost of constructing a transfer center as influencing factors, defines property of value for unit Euclidean distance as $C^*$, and the Euclidean distance with the property of value can be expressed as:

$$d(x^*, y^*) = C^* \cdot \sqrt{\sum_{i=1}^{q^*} (x_i^* - y_i^*)^2} = \frac{\sum_{i \in V} \sum_{j \in N} C \cdot q_i \cdot L_{ij} \cdot \beta_{ij} + \sum_{g \in G} Z_g \cdot \alpha_g}{\sum_{i \in V} \sum_{j \in N} L_{ij} \cdot \beta_{ij}} \sqrt{\sum_{i=1}^{q^*} (x_i^* - y_i^*)^2}$$

(30)

**5.1.2 Steps for solving the model.** Firstly, this paper provides the following definition:

1. If a production point $i$ ($i \in V$) has not yet established a transportation relationship with any transfer point, then the production point $i$ will be included in the set $A$ to be allocated.

2. If a production point $i$ terminates the transportation relationship with transfer point $g$ ($g \in G$), then $i$ will be included in the forbidden set $\bar{R}$ of transfer point g. Production point i is unable to establish a new transportation relationship with transfer point $g$.

At this point, the solving steps of the algorithm are as A1.

---

`Algorithm A1:`

`Step1: The initial capacity of the transfer point set is 0, clear the forbidden set` $\bar{R}_g$ `of transfer point g` ($g \in G$)`, and put all production points into the set A to be allocated.`

---

```
Step2: Perform cluster analysis on the production points in set A, use the minimum envelope method
to determine the position of transfer point g, and establish the transportation relationship
between production point i and transfer point g.
Step3: If ∑qᵢ ≥ Zg,the transportation relationships between the production points and the transfer
point is terminated in descending.
Step4: Put the production points that have been terminated transportation relationships into the set
A to be allocated and into forbidden set R̄ of transfer point g.
```

## 5.2 Genetic algorithm

The Genetic Algorithm was first proposed by John Holland from the United States in the 1970s, and is designed based on the laws of biological evolution. Genetic Algorithm is a computational model that simulates the natural selection and genetic theories of Darwin's theory of biological evolution. It is a method of searching for the optimal solution by simulating the process of natural evolution, which can quickly obtain results when solving more complex combinatorial optimization problems.

The genetic algorithm uses natural number encoding to randomly generate an initial population, and the specific calculation stepsof the algorithm are as A2:

```
Algorithm A2:

Step1: Set the population size inn, maximum number of iteration maxgen, crossover probability P_c, and
mutation probability P_m.
Step2: Obtain the initial population through clustering, calculate the fitness function f(x) based on
the objective function, and determine the current algebra n.
Step3: Follow the roulette wheel principle to select excellent individuals for inheritance to the
next generation.
Step4: Set the crossover probability P_c and mutation probability P_m, obtain a new population and
calculate the population fitness.
Step5: Perform a taboo search on the chromosomes, using the generated initial neighborhood solution
as the candidate solution set. Update the taboo table for taboo iteration to find the optimal solu-
tion, Current Algebra+1.
Step6: If the current algebra n = maxgen, and the calculation stops, the optimal solution for the
fitness function is the satisfactory solution to the problem [22].
```

## 5.3 Improved NSGA-II

From experience, it is known that the local search strategy, genetic rules, and mutation rules of basic genetic algorithm have a significant impact on the quality and computation time of solutions when solving route problems. Therefore, this paper focuses on the shortcomings of basic genetic algorithms, such as delayed information feedback, slow search speed, and strong dependence on the initial population, made improvements in genetic coding, fitness calculation, and proposed NSGA-II.. This algorithm uses non-dominated sorting to reduce the complexity of the algorithm, and uses crowding distance and crowding distance comparison operators to ensure the diversity of the population, accelerating the convergence speed of this algorithm. And use Matlab to write the program, ultimately designing an improved NSGA-II [23].

(1) Initialize the population

The quality of initial feasible solutions will directly affect the efficiency of swarm intelligence optimization algorithms. In NSGA-II, a random generation method is used to generate the initial population, which may lead to insufficient population diversity and low convergence accuracy of the algorithm. To overcome this deficiency, this paper adopts the following improvement method to generate the initial population:

Step 1: Randomly generate a temporary population with a size of $2 \times inn$, use three objective functions $f_1, f_2, f_3$ as indicators to calculate the spatial distance $w_i$ between each chromosome and other chromosomes, and sort $w_i$ accordingly.

$$w_i = \sum_{j,i \neq j} (|f_1^i - f_1^j| + |f_2^i - f_2^j| + |f_3^i - f_3^j|)$$

(31)

Step 2: Select $inn$ chromosomes from $w_i$ in descending order to form the initial population.

(2) The selection strategy of Pareto solution

In response to the difficulty in determining the weights of indicators in traditional multi-objective optimization problems, this paper uses Pareto solution to balance the importance of multiple objectives. Pareto solution refers to a solution that cannot be improved on one objective without being inferior to other objectives under given resource conditions. In other words, if there is a change that makes at least one person better without making anyone worse off, then this change cannot further improve the solutions of other objective functions, which constitute the Pareto solution set. Its definition is as follows:

Pareto solution is a balance point found between multiple objectives, where the improvement of any one objective does not compromise the performance of the others.

Definition 3: All Pareto solutions to form a set $U^*$, where the cost and risk of plan $i \in U^*$ are denoted as $cost(i)$ and $risk(i)$, respectively;

Definition 4: The weighted average values of cost and risk for all transportation route schemes as $\overline{cost}$ and $\overline{risk}$, respectively.

The selection strategy for Pareto solutions is as follows

① If there is a unique scheme $i \in U^*$ that satisfies the minimum $|risk(i) - \overline{risk}|$, then plan $i$ is chosen as the Pareto solution.

② If there is scheme $i, j \in U^*$ that satisfies the minimum $|risk(i) - \overline{risk}| = |risk(j) - \overline{risk}|$ and $|risk(i) - \overline{risk}|$, $|cost(i) - \overline{cost}| < |cost(j) - \overline{cost}|$, then plan $i$ is chosen as the Pareto solution.

(3) If there is scheme $i, j \in U^*$ that satisfies the minimum $|risk(i) - \overline{risk}| = |risk(j) - \overline{risk}|$ and $|risk(i) - \overline{risk}|$, $|cost(i) - \overline{cost}| = |cost(j) - \overline{cost}|$, then when $risk(i) < risk(j)$, plan $i$ is chosen as the Pareto solution.

(3) Fast non-dominated sorting

The concept of domination: If individual A is not inferior to another individual B in all goals and is superior to individual B in at least one goal, it is considered that A dominates B.

The fast non-dominated sorting algorithm is generally used to solve multi-objective optimization problems, by determining the non-dominated level of each individual in the population, sorting them in order to find the optimal solution set. This algorithm can effectively perform non-dominated sorting on individuals in the population without using additional space, and is suitable for handling populations with a large number of individuals. The specific sorting method is:

Individual $i$ in the population has two attributes: $n(i)$ (the number of individuals that dominate it) and $S(i)$ (the number of individuals that are dominated by it). When $n(i) = 0$, it indicates that individual $i$ is not dominated by any other individual and assign individual $i$ to the first non dominated layer $F(1)$. Then, define the set of individuals dominated by individual i as $K$, and subtract 1 from the $n(K)$ of all individuals in set $K$, and so on, until all individuals are graded.

(4) Crowding Distance

In NSGA-II, crowding distance is an important criterion for evaluating individual strengths and weaknesses. Crowding distance=(individual fitness – average fitness)/standard deviation. Crowding distance can ensure that the algorithm converges to a relatively uniform Pareto domain, achieving the goal of maintaining population diversity.

At this point, drawing on the solution approach of multi-objective optimization algorithms [24–26], the calculation process of NSGA-II is designed as Fig 4.

## 6 Numerical experiments

### 6.1 Basic experiments for an idealized district

Taking Lingang University City in Cuiping District, Yibin City as an example. there are a total of 58 medical waste generation points, 4 transfer points, and 1 waste disposal center in the area shown in Fig 5. Determine two echelons of infectious medical waste transportation plans with the goal of minimizing the total cost of the transportation plan.

In Fig 5, yellow areas such as 1, 2,... represent the point of generation of infectious medical waste, red areas such as A, B,... represent the medical waste transfer center, and blue square O represents the disposal point of infectious medical waste.

Table 3 shows the node coordinates and demand. Based on GIS coordinates, statistics are conducted in km units, with one decimal place retained. The data before and after production"/" respectively represent the amount of medical waste produced from nodes 1 to 58 before and after the outbreak of the epidemic.

The quantities of A, B, C, D (transfer points)and O(disposal center) represent the maximum capacity of waste at transfer points and disposal center before and after the outbreak of the epidemic.

Considering the natural isolation formed by the terrain and the complex road connectivity between nodes, this paper converts some distance into a straight line distance between node coordinates. The distance between nodes in the obstacle region is denoted as $\infty$.

Due to the significant increase in the amount of infectious medical waste during the COVID-19 pandemic, the volume of such waste is expected to rise substantially, leading to the capacity of existing transfer centers being unable to meet the demand of the regional system. As a result, additional transfer centers need to be established. Meanwhile, due to the sharp rise in the volume and frequency of infectious medical waste transportation, it is necessary to deploy a certain number of rescue centers within the region to ensure that the coverage area of the rescue centers fully encompasses the Lingang University Town area. Furthermore, given the uncertainty in the measurement of transportation risks, the remaining relevant parameters are set as shown in Table 4 to simplify the model in this paper.

Due to the significant increase in the amount of medical waste produced during the epidemic, the capacity of transfer points is difficult to meet the regional transfer needs. Therefore, it is necessary to add several transfer points. At the same time, due to the increase in the volume and frequency of medical waste transportation, it is also required to set up a certain number of rescue centers in the area to ensure that the rescue scope of the rescue centers can fully cover the Lingang University City area. Assuming that the cost of adding transfer points and rescue centers is both M, and the coverage radius of the rescue center is 5 km.

### 6.2 Transportation plan for medical waste with low production volume

The parameter values in the algorithm are: genetic population size $inn = 100$, genetic crossover probability $P_c = 0.7$, genetic mutation probability $P_m = 0.05$, and termination of algebraic generation $\max gen = 200$.

The improved NSGA-II is used to solve the example, and the algorithm terminates after 200 iterations. The Pareto solution obtained is the optimal route organization scheme for the problem. Due to the relatively small amount of medical waste generated at this point, only a small number of vehicles are needed for transportation. Each vehicle has a longer transportation time and a low full load rate, so there is no need to build a new transfer point. By using Matlab to solve the problem, the average distance of each Pareto optimal solution tends to stabilize after about 160 iterations, the average distance between each Pareto optimal solution tends to stabilize. The Pareto front is shown in Fig 6. According to the Pareto solution selection strategy, the recommended path for medical waste recycling is shown in Fig 7 and Table 5.

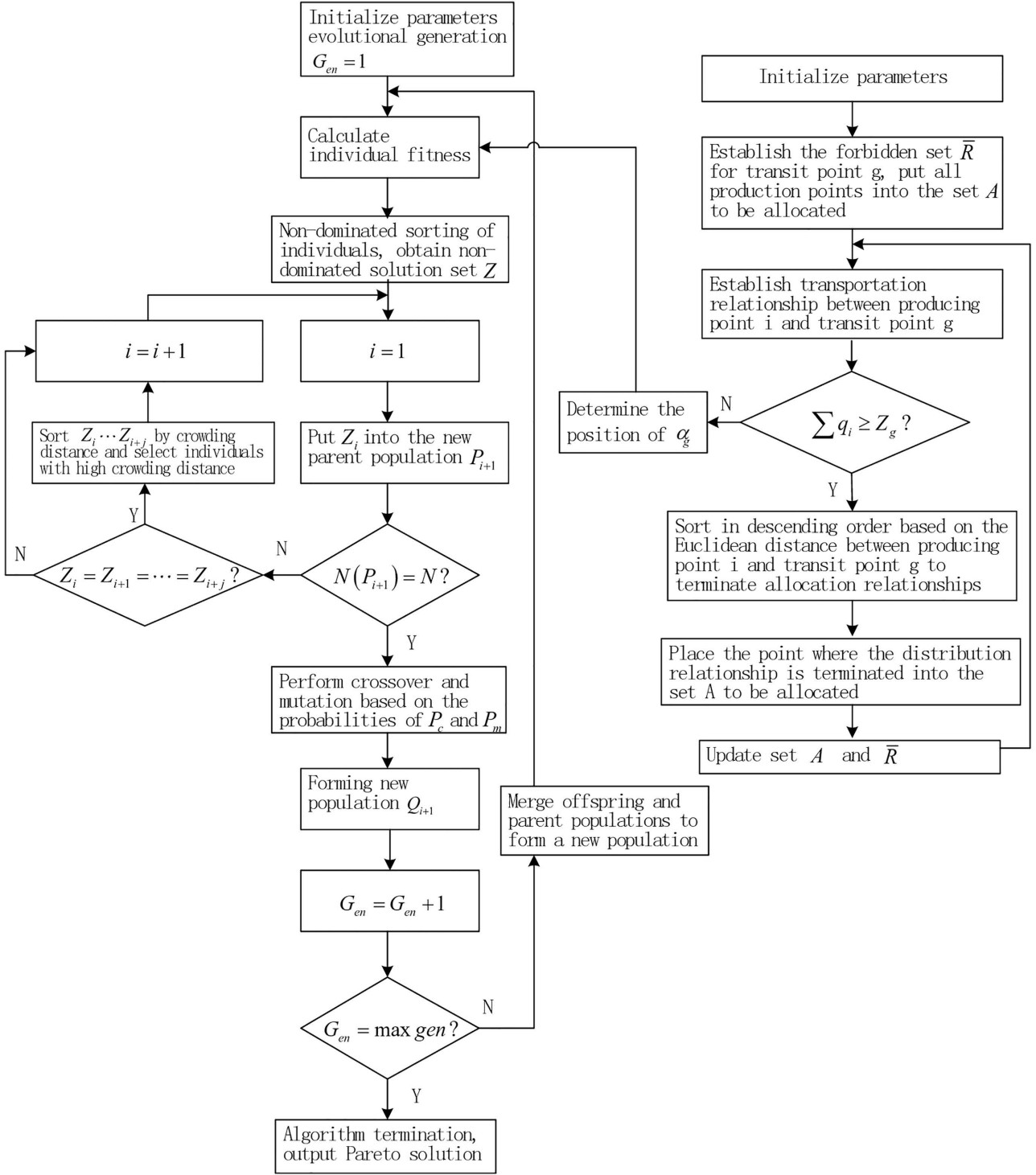

**Fig 4. NSGA-II process based on minimum envelope clustering.**

---

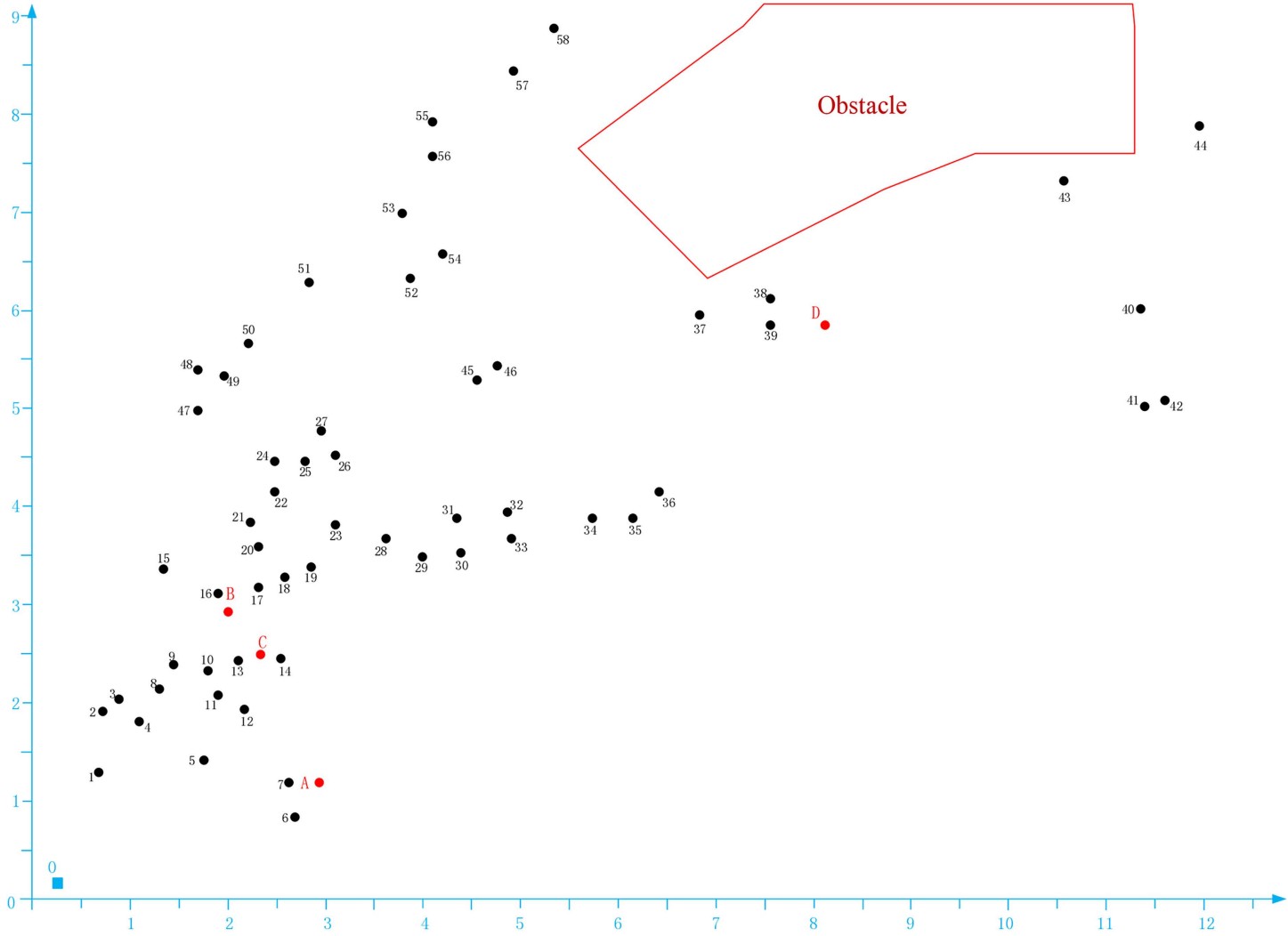

**Fig 5. Distribution of generation points, transfer points, and disposal centers for medical waste.**

At this point, considering the small amount of medical waste generated, there is no need to set up rescue points and the transportation risk objective function (11) is not considered.

### 6.3 Transportation routes when the amount of medical waste produced increases significantly

When the amount of medical waste generated increases significantly, such as during the epidemic, the waste exceeds the capacity of transfer points, so it is necessary to build several new transfer points. At the same time, the frequent transportation of medical waste will lead to an increase in transportation risks, and it is necessary to build a certain number of rescue centers.

Based on the data from Table 2 after the outbreak of the epidemic, two new transfer points, E and F, with coordinates E (0.9, 1.9) and F (4.6, 3.8), were constructed, with capacities of 2200 kg and 2500 kg, respectively. By using Matlab to solve the problem, the average distance of each Pareto optimal solution tends to stabilize after about 170 iterations, the average distance between each Pareto optimal solution tends to stabilize. The Pareto front is shown in Fig 8. According to the Pareto solution selection strategy, the recommended path for medical waste recycling is shown in Fig 9 and Table 6.

**Table 3. Node coordinates and production.**

| Node | X-axis | Y-axis | Production(t) | Node | X-axis | Y-axis | Production(t) | Node | X-axis | Y-axis | Production(t) |
|------|--------|--------|---------------|------|--------|--------|---------------|------|--------|--------|---------------|
| O | 0.5 | 0.4 | $\infty$ /$\infty$ | 17 | 2.3 | 3.2 | 38/255 | 38 | 7.5 | 6.2 | 36/181 |
| A | 2.8 | 1.2 | 2500 | 18 | 2.6 | 3.3 | 34/257 | 39 | 7.5 | 5.8 | 47/253 |
| B | 2.0 | 2.9 | 3200 | 19 | 1.8 | 3.4 | 26/296 | 40 | 11.4 | 6.0 | 33/144 |
| C | 2.3 | 2.5 | 2000 | 20 | 2.3 | 3.6 | 31/142 | 41 | 11.4 | 5.0 | 43/150 |
| D | 8.1 | 5.8 | 3000 | 21 | 2.2 | 3.8 | 33/305 | 42 | 11.6 | 5.1 | 33/246 |
| 1 | 0.7 | 1.3 | 31/259 | 22 | 2.5 | 4.2 | 46/154 | 43 | 10.6 | 7.3 | 27/194 |
| 2 | 0.7 | 1.9 | 40/232 | 23 | 3.1 | 3.8 | 40/262 | 44 | 11.9 | 7.9 | 38/259 |
| 3 | 0.8 | 2.1 | 42/293 | 24 | 2.5 | 4.5 | 46/313 | 45 | 4.5 | 5.3 | 50/232 |
| 4 | 1.1 | 1.8 | 43/252 | 25 | 2.8 | 4.5 | 38/159 | 46 | 4.7 | 5.4 | 46/257 |
| 5 | 1.7 | 1.4 | 31/267 | 26 | 3.4 | 4.5 | 38/321 | 47 | 1.6 | 5.0 | 49/216 |
| 6 | 2.7 | 0.8 | 47/297 | 27 | 2.9 | 4.8 | 43/202 | 48 | 1.6 | 5.4 | 44/154 |
| 7 | 2.6 | 1.2 | 27/154 | 28 | 3.6 | 3.7 | 41/179 | 49 | 1.9 | 5.3 | 39/174 |
| 8 | 1.3 | 2.2 | 45/278 | 29 | 4.0 | 3.5 | 48/150 | 50 | 2.2 | 5.7 | 27/309 |
| 9 | 1.4 | 2.4 | 37/215 | 30 | 4.4 | 3.5 | 34/277 | 51 | 2.8 | 6.3 | 37/141 |
| 10 | 1.8 | 2.3 | 29/265 | 31 | 4.3 | 3.8 | 36/310 | 52 | 3.9 | 6.3 | 44/305 |
| 11 | 1.9 | 2.1 | 37/191 | 32 | 4.8 | 3.9 | 50/188 | 53 | 3.7 | 7.0 | 37/307 |
| 12 | 2.2 | 1.9 | 31/159 | 33 | 4.9 | 3.7 | 45/237 | 54 | 4.2 | 6.6 | 40/291 |
| 13 | 2.1 | 2.4 | 34/193 | 34 | 5.7 | 3.8 | 29/132 | 55 | 4.1 | 7.9 | 29/248 |
| 14 | 2.5 | 2.5 | 27/276 | 35 | 6.1 | 3.8 | 45/187 | 56 | 4.1 | 7.6 | 39/249 |
| 15 | 1.3 | 3.4 | 43/225 | 36 | 6.4 | 4.2 | 35/147 | 57 | 4.9 | 8.4 | 28/264 |
| 16 | 1.9 | 3.1 | 43/302 | 37 | 6.8 | 5.9 | 46/322 | 58 | 5.4 | 8.8 | 34/314 |

In addition, due to the significant increase in waste generation for the instance, E and F are new transfer points. With the goal of fully covering the area and minimizing transportation risks, the newly built rescue centers are located at points B and D (also transfer points), which can fully cover all nodes, with a construction cost of 200 million yuan. It can be seen that when the amount of waste produced by each node suddenly increases, the total mileage of the line does not change much and shows a fluctuating growth. The number of vehicles will experience a significant stepwise increase. Overall, transportation costs are showing an upward trend, but with the increasing number of vehicles and the construction of new transfer and rescue centers, transportation costs will also experience a fluctuating ladder in the upward trend.

## 6.4 Comparison and analysis

To further validate the performance of the improved NSGA-II algorithm and the model, a comparison was made between the improved NSGA-II algorithm and the traditional genetic algorithm. The initial population, genetic probability, mutation probability, and termination algebra parameters of each system are set the same. At the same time, considering the actual situation during the period, the amount of medical waste produced will fluctuate with government policies, the spread of the epidemic, and other factors. The plans will be operated separately under uncertain demand conditions, and the comparison of the results is shown in Table 7.

From Table 7, it can be seen that the improved NSGA-II algorithm designed in this paper achieves satisfactory calculation results in different situations. Therefore, by utilizing the model and algorithm presented in this paper, the optimal transfer point location and vehicle transportation plan for medical waste can be effectively obtained in various situations.

In addition, we calculated the hyper volumes of each result, as shown in Fig 10. Hyper volume is an important parameter reflecting the Pareto front quality. It can be seen that the improved NSGA-II has the highest hyper volume in both of the previous examples, and its results are better than those of the basic genetic algorithm.

**Table 4. Explanation and Values of Relevant Parameters in the Case.**

| Symbols | Parameter | Values |
|---|---|---|
| $\mu$ | Average wind speed | 4.43km/h |
| $\theta$ | Probability of waste being infectious | 0.6 |
| $P_{ij}$ | The population density of all lines in the region | 7500per/km$^2$ |
| $H$ | Vehicle height | 1 meter |
| $FC_g$ | Cost of adding rescue centers | 50 million yuan |
| $F_D$ | The rescue capacity of the rescue center | 1000 kg/h |
| $R_D$ | The coverage radius of the rescue center | 500 meters |
| $v_D$ | Driving speed | 40 km/h |
| $Z_{k,1}$ | Vehicle load capacity in first echelon | 5 t |
| $Z_{k,2}$ | Vehicle load capacity in second echelon | 1 t |
| $FC_{k,1}$ | Fixed cost in first echelon | 5million yuan |
| $FC_{k,2}$ | Fixed cost in second echelon | 2million yuan |
| $fc_{k,1}$ | Unit mileage transportation costin first echelon | 100 thousand yuan |
| $fc_{k,2}$ | Unit mileage transportation costin second echelon | 50 thousand yuan |

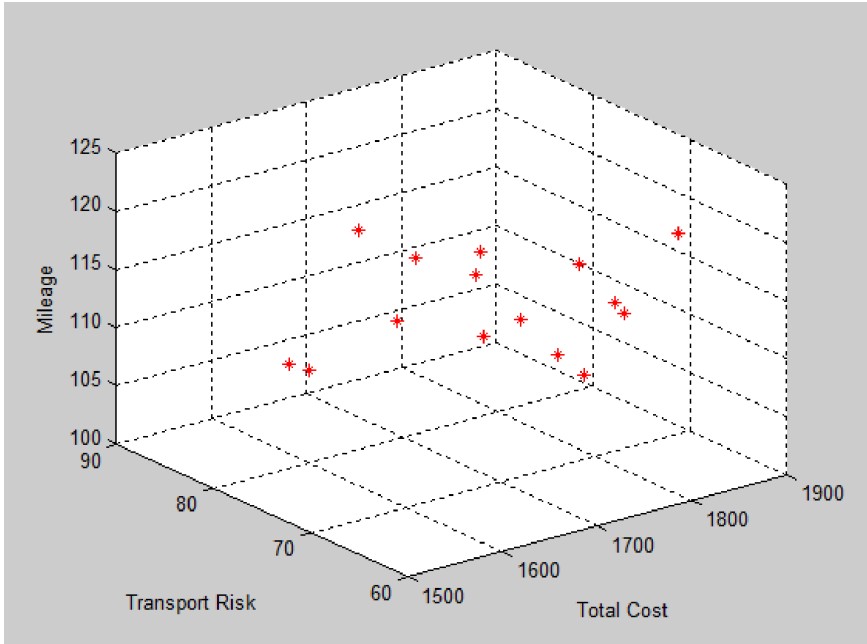

**Fig 6. Solution obtained by NSGA-II (Instance #1).**

In summary, the feasible solutions obtained through the improved NSGA-II perform better in solving large-scale transportation problems. At the same time, simplifying the number of nodes, optimizing node demand, and removing obstacles can all reduce computation time and workload. Overall, when using the improved NSGA-II to calculate the site selection and transportation route of medical waste transfer points, the calculation process and obtained transportation plan are reasonable.

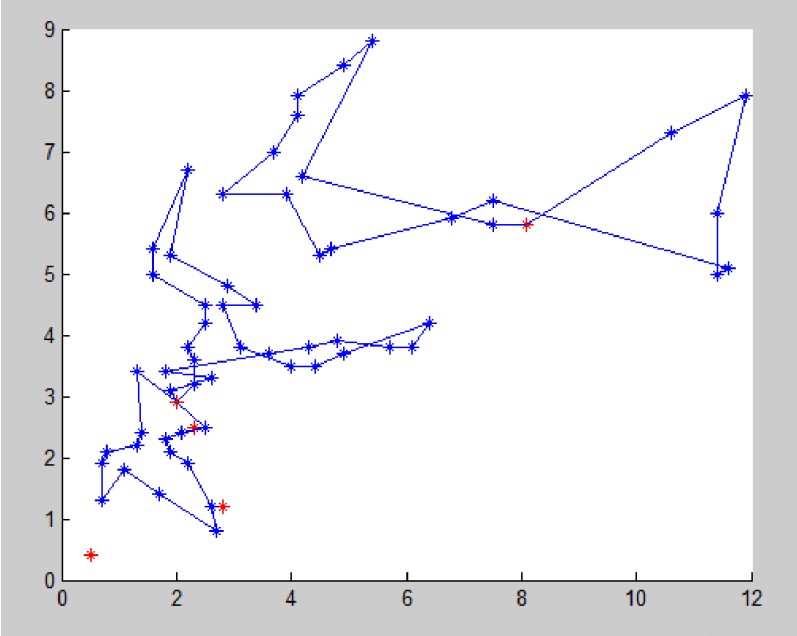

**Fig 7. Vehicle transportation plan.**

**Table 5. Delivery plan results for low medical waste production.**

| Optimal route | Path | Total Cost | Transport risk | Vehicle mileage | Vehicle Number |
|---|---|---|---|---|---|
| **First Stage** | Path1_1<br>Path1_2<br>Path1_3 | 1063 | 73.12 | 92.62 | 3 |
| **Second Stage** | Path2_1 | 690 | 0 | 18.99 | 1 |
| **Total** | | 1753 | 73.12 | 111.61 | 4 |

Path1_1 represents: B-17-20-21-22-24-47-48-50-49-27-26-25-23-29-30-33-36-

35-34-32-31-28-19-18-16-B;

Path1_2 represents: B-14-13-10-11-12-7-6-5-4-1-2-3-8-9-15-B;

Path1_3 represents: D-43-44-40-41-42-38-37-46-45-52-51-53-56-55-57-58-54-39-D;

Path2_1 represents: O-B-D-O.

## 7 Conclusions

The problem studied in this paper is the collection, transportation, transfer, and disposal of infectious medical waste in cities. By determining the location of transfer points and optimizing the paths between various nodes (production points, transfer points, and disposal centers), a two-echelon multi-objective route optimization model is constructed to achieve the goal of minimizing the total cost and transportation risk of the entire waste transportation process. To solve the model, an improved NSGA-II based on minimum envelope clustering was designed:

(1) Using chromosome encoding composed of numerical strings to represent transportation routes and generate initial populations.

(2) Adopting the rules of an improved genetic algorithm, the steps of crossover, mutation, taboo table, etc. are improved to adjust convergence speed and enhance optimization ability.

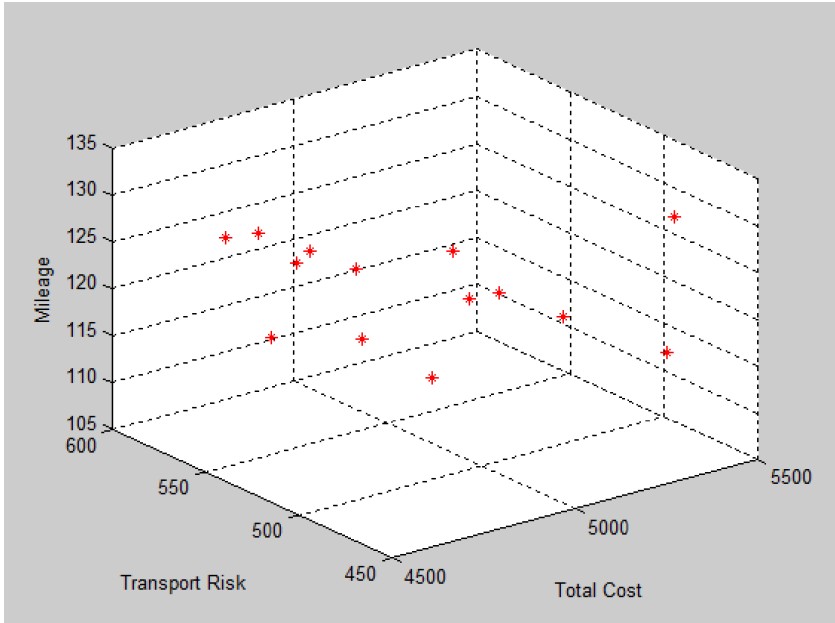

**Fig 8. Solution obtained by NSGA-II (Instance #2).**

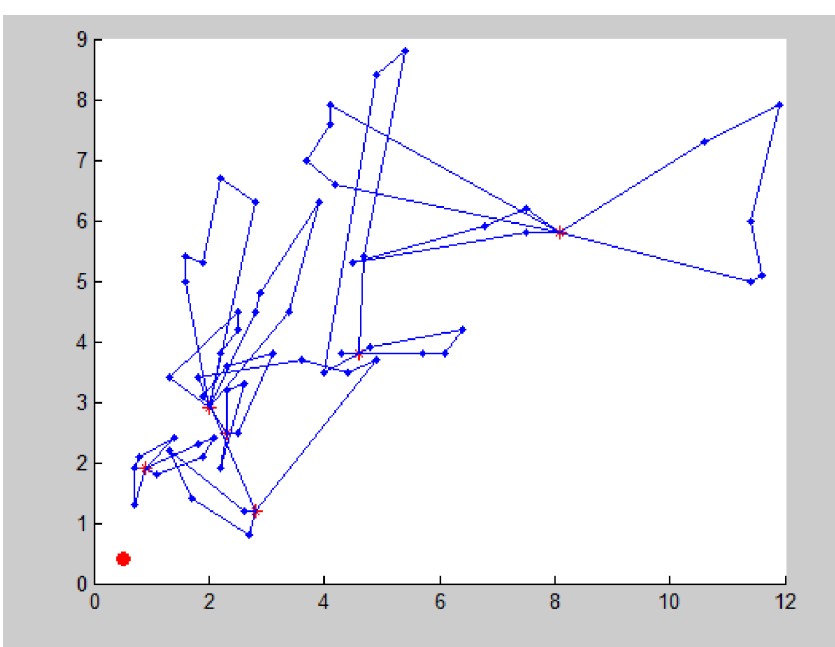

**Fig 9. Transportation plan when the amount of medical waste generated increases significantly.**

**Table 6. Delivery plan results for increased medical waste production.**

| Optimal route | Path | Total Cost | Transport risk | Vehicle mileage | Vehicle Number |
|---|---|---|---|---|---|
| **First Stage** | A-7-8-5-6-A<br>A-19-28-30-33-A<br>B-15-24-22-21-B<br>B-47-48-49-50-51-B<br>B-25-27-52-26-B<br>C-16-20-23-14-C<br>C-17-18-12-C<br>D-55-56-53-54-D<br>D-43-44-40-42-41-D<br>D-38-37-45-39-D<br>E-9-3-2-1-E<br>E-10-13-11-4-E<br>F-32-36-35-34-31-F<br>F-46-58-57-29-F | 3249 | 517.26 | 89.78 | 14 |
| **Second Stage** | O-F-D-O<br>O-C-B-O<br>O-A-E-O | 1809 | 0 | 30.87 | 3 |
| **Total** | | 5058 | 517.26 | 120.65 | 17 |

**Table 7. Comparison of calculation results.**

| | Improved NSGA-II | | MOEA/D | | Traditional genetic algorithm | | Average optimization rate |
|---|---|---|---|---|---|---|---|
| Objective function | low waste production | increased waste production | low waste production | | increased waste production | low waste production | increased waste production |
| The amount of medical waste produced(kg) | 2209 | 13435 | 2209 | 13435 | 2209 | 13435 | / |
| Vehicle driving distance(km) | 111.61 | 120.65 | 111.61 | 130.24 | 120.23 | 134.58 | 8.76% |
| Number of vehicles | 4 | 17 | 4 | 18 | 5 | 19 | 12.78% |
| Total transportation cost(yuan) | 1753 | 5058 | 1753 | 5303 | 2037 | 5589 | 11.72% |
| Transportation risk(t·person/km$^3$) | – | 517.26 | – | 520.58 | – | 527.97 | 2.03% |
| The full load rate of the vehicle | 55.23% | 92.66% | 55.23% | 90.12% | 49.09% | 86.68% | 6.06% |

(3) Comprehensively calculate the fitness of individuals based on non dominated sorting and crowding distance.

(4) Repeat the steps to determine the Pareto solution set.

To verify the ability of the algorithm, actual cases were selected for calculation, and the optimal paths were considered for different amounts of waste generated. By comparing with traditional genetic algorithms, it was found that the improved NSGA-II in this paper performs better, both when the amount of medical waste generated is low and after a significant increase: the average total transportation mileage is reduced by 8.767%, the average number of transportation vehicles is reduced by 12.78%, the average transportation cost is reduced by 11.72%, the average transportation risk is reduced by 2.03%, and the average full load rate is increased by 6.06%. The case results show that the algorithm quickly converges to the vicinity of the maximum value in the early stage and performs stably in cases of different scales. So the algorithm has good convergence and stability when solving complex route optimization problems. The model and algorithm presented in this paper have broad applicability, providing disposal solutions for the site selection and path optimization of infectious medical waste recycling, as well as ideas for other types of waste disposal.

However, this paper also has limitations, such as the lack of specialized qualifications and facility equipment requirements for the treatment of infectious medical waste in the study, and the possibility of irregular transportation strategies

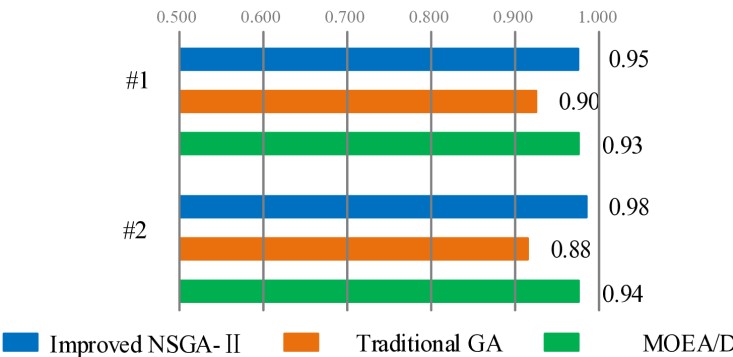

**Fig 10. Hypervolumes of each result.**

for transfer points that generate less medical waste. At the same time, this article did not consider the chemical and risk characteristics of different types of medical waste, and urban medical waste needs to be classified and treated according to different characteristics. Future research can be expanded from the following aspects: (1) increasing sensitivity analysis. In cost sensitivity analysis, adding uncertain demand to the expected number of vehicles, fuel costs under different load conditions of vehicles, and introducing the time-varying characteristics of road traffic in traffic environment sensitivity to affect transportation risks and rescue efficiency. (2) Increasing the uncertainty of urban road traffic and its impact on the travel paths of medical waste recycling vehicles, such as considering the dynamic changes in real-time transportation conditions of urban roads in the calculation of rescue speed objectives. (3) Including the dynamic generation rate of medical waste and its impact on cost objectives, such as factoring in the vehicle idle cost coefficient under different medical waste generation rates in the fixed vehicle usage costs. (4) Considering the risk characteristics of different types of medical waste and the impact of climate conditions in different geographical areas on the spread of risks, as well as regulatory constraints and compliance issues related to infectious waste logistics. Additionally, extending the model to include other types of medical waste or different geographical regions. At the same time, emerging technologies such as machine learning and artificial intelligence can be integrated, such as combining big data technology to analyze the impact of time-varying weather conditions and road traffic conditions on the risk of medical waste transportation, and combining deep reinforcement learning technology to optimize vehicle routes.

## Supporting information

**S1 Fig. This is The amount of medical waste generated and processed in China.**
(TIF)

**S2 Fig. This is Infectious medical wastes transport system.**
(TIF)

**S3 Fig. This is Schematic diagram of virus spread range.**
(TIF)

**S4 Fig. This is Symbols used in the paper.**
(TIF)

**S5 Fig. This is Distribution of generation points, transfer points, and disposal centers for medical waste.**
(TIF)

**S6 Fig. This is Solution obtained by NSGA-II (Instance #1).**
(TIF)

**S7 Fig. This is Vehicle transportation plan.**
(TIF)

**S8 Fig. This is Solution obtained by NSGA-II (Instance #2).**
(TIF)

**S9 Fig. This is Transportation plan when the amount of medical waste generated increases significantly.**
(TIF)

**S10 Fig. This is Hypervolumes of each result.**
(TIF)

**S1 Table. This is Comparison ofresearch contributions withexisting studies.**
(DOCX)

**S2 Table. This is Symbols used in the paper.**
(DOCX)

**S3 Table. This is Node coordinates and production.**
(DOCX)

**S4 Table. This is Explanation and Values of Relevant Parameters in the Case.**
(DOCX)

**S5 Table. This is Delivery plan results for low medical waste production.**
(DOCX)

**S6 Table. This is Delivery plan results for increased medical waste production.**
(DOCX)

**S7 Table. This is Comparison of calculation results.**
(DOCX)

## Author contributions

**Conceptualization:** Libo LI.

**Data curation:** Libo LI.

**Formal analysis:** Xuerui QIN.

**Methodology:** hao Chen.

**Project administration:** hao Chen.

**Resources:** hao Chen.

**Software:** hao Chen.

**Validation:** Wenxian Wang.

**Writing – original draft:** hao Chen.

**Writing – review & editing:** hao Chen, Libo LI, Xuerui QIN, Wenxian Wang.

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
