## [Decision Letter · Decision Letter 0]

16 Dec 2024

Dear Dr. Chen,

Thank you for submitting your manuscript to PLOS ONE. After careful consideration, we feel that it has merit but does not fully meet PLOS ONE’s publication criteria as it currently stands. Therefore, we invite you to submit a revised version of the manuscript that addresses the points raised during the review process.

We look forward to receiving your revised manuscript.

Kind regards,

Erfan Babaee Tirkolaee, PhD

Academic Editor

PLOS ONE

Journal Requirements:

The first author would like to express gratitude for the support from Yibin University

(No.412-2022QH23) and Sichuan Provincial Key Laboratory of Automobile Measurement and

Control and Safety (No.QCCK2019-006) and Yibin Federation of Social Science Associations (No.

2024YBSKL81).

5. Please ensure that you refer to Figure 1 in your text as, if accepted, production will need this reference to link the reader to the figure.

6. We note that Figure 4 in your submission contain [map/satellite] images which may be copyrighted. All PLOS content is published under the Creative Commons Attribution License (CC BY 4.0), which means that the manuscript, images, and Supporting Information files will be freely available online, and any third party is permitted to access, download, copy, distribute, and use these materials in any way, even commercially, with proper attribution. For these reasons, we cannot publish previously copyrighted maps or satellite images created using proprietary data, such as Google software (Google Maps, Street View, and Earth). For more information, see our copyright guidelines: http://journals.plos.org/plosone/s/licenses-and-copyright.

a. You may seek permission from the original copyright holder of Figure 4 to publish the content specifically under the CC BY 4.0 license.  

7. Please remove your figures from within your manuscript file, leaving only the individual TIFF/EPS image files, uploaded separately. These will be automatically included in the reviewers’ PDF.

Reviewers' comments:

Reviewer's Responses to Questions

**Comments to the Author**

1. Is the manuscript technically sound, and do the data support the conclusions?

Reviewer #1: Yes

Reviewer #2: Yes

2. Has the statistical analysis been performed appropriately and rigorously?

Reviewer #1: Yes

Reviewer #2: Yes

3. Have the authors made all data underlying the findings in their manuscript fully available?

Reviewer #1: Yes

Reviewer #2: Yes

4. Is the manuscript presented in an intelligible fashion and written in standard English?

Reviewer #1: Yes

Reviewer #2: Yes

Reviewer #1: Thank you for your submission of the manuscript. Your study tackles an important and timely issue regarding the management of medical waste, especially during the COVID-19 pandemic. The aim of developing a multi-objective optimization model to minimize costs and risks is commendable and has the potential to contribute significantly to the field. However, there are several areas that require clarification and improvement to enhance the overall quality and comprehensibility of your manuscript.

1. Can you elaborate on why a two-echelon model was chosen over other potential models? What specific advantages does it offer in the context of medical waste transportation?

2. The paper mentions a three-dimensional risk measurement sub-model. Could you provide more details on how this model was constructed and the specific parameters it includes?

3. The manuscript states that the weight ratio of optimization objectives lacks a basis. Can you clarify how you determined the weights for transportation costs and risks in your model?

4. You mention a lack of stability and sensitivity analysis. Could you provide a plan for how you intend to conduct this analysis in future work?

5. In the NSGA-II algorithm, how is the initial population generated? What criteria are used to ensure diversity in the initial solutions?

6. Ensure that all references are up-to-date and formatted according to the journal's guidelines. Some citations appear to be missing or incomplete. In general, the authors are recommended to refer to more recent papers in the manuscript, such as: doi.org/10.1016/j.eswa.2023.121035, doi.org/10.1109/TITS.2023.3285430.

7. The case study in Yibin City is mentioned, but details on the data used (e.g., waste generation rates, vehicle capacities) are sparse.

8. What assumptions did you make in your model regarding vehicle capacities, waste generation rates, and transit point locations? How might these assumptions impact the results?

9. Some figures and tables lack clear explanations in the text. Can you ensure that all visual aids are adequately referenced and explained?

10. What are the limitations of your study, and how might they affect the applicability of your model in different contexts or regions?

11. What are the next steps for this research? Are there plans to extend the model to include other types of medical waste or different geographical areas?

I look forward to seeing the revised manuscript that addresses these concerns.

Reviewer #2: This paper use NGGA-II to find the optimum transportation routes for infectious medical waste. The problem of this work is interesting because this optimization problem due with many parameters. However, this paper msut be improve in some point to increase the quality of this work. My comments are below:

1) I reccommend the author to change the description of objective f_3 from section 4.4 to 4.2.3. It very hard and find where is the description of objective 3. Because the 4.1.1 and 4.2. is describe about f_1 and f_2.

2) This paper use NGSA-II to find the 3 objectives optimization problem. However, I cannot see you Pareto solutions of your problem. I reccomend the author to added the figure of Pareto solutions in your manuscript.

3) The author must be implied why where is the selected optimum solution from the Pareto solutions and explained why this point is good for selected.

4) In general, the history search of multi/many-objective optimization can measured by multi-objective optimiztion metrics. I reccomend the author to used some metrics to shown your optimum solution has been converge.

5) This paper try to compared the results of improve NSGA-II and traditional NSGA-II. However, the comparison isn't fair because now you due with the multi-objective optimization. I reccommend the author compared the Pareto solution from each method and compared the convergence rate of each algorithm using some metric such as hypervolume.

**Do you want your identity to be public for this peer review?** For information about this choice, including consent withdrawal, please see our Privacy Policy

Reviewer #1: No

Reviewer #2: No

---

## [Author Response · Author response to Decision Letter 1]

12 Apr 2025

Original Manuscript ID: PONE-D-24-35149

Original Article Title: “Research on Optimization of Transportation Routes for Infectious Medical Waste”

To: Plos One Editor

Re: Response to reviewers

Dear Editor,

Thank you for allowing a resubmission of our manuscript, with an opportunity to address the reviewers’ comments.

We are uploading (a) our point-by-point response to the comments (below) (response to reviewers), (b) an updated manuscript with yellow highlighting indicating changes, and (c) a clean updated manuscript without highlights (Word main document).

Best regards,

Hao Chen, Libo Li, Xuerui Qin*, Wenxian Wang.

Reviewer 1 Comments to the Author

My comments are as follows:

1. Can you elaborate on why a two-echelon model was chosen over other potential models? What specific advantages does it offer in the context of medical waste transportation?

According to the reviewer’s requirement, relevant statements have been added to in Section 3.1.

" From Figure 1, it can be seen that the infectious medical waste transportation system is architecture in above the urban road network containing emergency system� including the produce points (medical institutions), transfer points, and disposal centers of medical waste. In the first stage, transportation vehicles depart from the transfer point, collect medical waste from several production points, and then return to the transfer point for temporary storage; The second stage is similar to the first stage, starting from the disposal center, passing through several transfer centers, and returning to the disposal center. If a leakage accident occurs during transportation, the rescue center in the area will quickly handle it to control the spread of the virus. Therefore, the problem of transfer point location - transportation route decision is a two-stage multi-objective optimization problem, that involves collaborative optimization of multiple links such as rescue centers, medical institutions, transfer points, and disposal centers in the event of a virus leak."

2. The paper mentions a three-dimensional risk measurement sub-model. Could you provide more details on how this model was constructed and the specific parameters it includes?

According to the reviewer’s requirement, relevant statements have been added to in Section 3.2.

“It can be defined that vehicles transporting infectious medical waste are risk points, and the trajectory of the risk point is a line segment. Considering the characteristics of infectious viruses, their diffusion process can be described as follows: starting from vehicles, the virus spreads in a three-dimensional manner with a certain diffusion radius. Within a certain period of time, the virus will roughly form a region similar to a half cylinder, as shown in Figure 2. It can be found that the risk of virus transmission mainly depends on parameters such as spread radius , transportation path length , and vehicle height .”

According to the reviewer’s requirement, relevant statements have been added to in Section 4.2.2.

“The total transportation risk is represented by the product of transportation risk, diffusion time, and population density along each section of the route.

The radius of the diffusion range is:

5

Among them, the diffusion time can be expressed as:

6

The cross-sectional area of the diffusion range is:

7

is the length of the route, and the volume of diffusion range can be expressed as:

8

The concentration of the virus is represented by the population within the current diffusion range:

9

Based on equations 4) and (5), the transportation risk on the route can be expressed as:

10

Transportation risk is represented by:

11�.”

3. The manuscript states that the weight ratio of optimization objectives lacks a basis. Can you clarify how you determined the weights for transportation costs and risks in your model?

According to the reviewer’s requirement, relevant statements have been added to in Section 2.

“Therefore, based on the environmental transmission characteristics of viruses, this paper proposes a multi-objective, three-dimensional risk measurement model that considers the risk of virus transmission and the generalized system cost. Moreover, this paper introduces the impact of road traffic volume on the risk of virus transmission, and constructs a model of transfer point locations - transportation routes with the goal of minimizing total system cost and transmission risk. Finally, Pareto solution is obtained based on minimum envelope clustering and NSGA-II.”

According to the reviewer’s requirement, relevant statements have been added to in Section 5.3.

“From experience, it is known that the local search strategy, genetic rules, and mutation rules of basic genetic algorithm have a significant impact on the quality and computation time of solutions when solving route problems. Therefore, this paper focuses on the shortcomings of basic genetic algorithms, such as delayed information feedback, slow search speed, and strong dependence on the initial population, made improvements in genetic coding, fitness calculation, and proposed NSGA-II.. This algorithm uses non-dominated sorting to reduce the complexity of the algorithm, and uses crowding distance and crowding distance comparison operators to ensure the diversity of the population, accelerating the convergence speed of this algorithm. And use Matlab to write the program, ultimately designing an improved NSGA-II [24-25].

(2) The selection strategy of Pareto solution

In response to the difficulty in determining the weights of indicators in traditional multi-objective optimization problems, this paper uses Pareto solution to balance the importance of multiple objectives. Pareto solution refers to a solution that cannot be improved on one objective without being inferior to other objectives under given resource conditions. In other words, if there is a change that makes at least one person better without making anyone worse off, then this change cannot further improve the solutions of other objective functions, which constitute the Pareto solution set.”

4. You mention a lack of stability and sensitivity analysis. Could you provide a plan for how you intend to conduct this analysis in future work?

According to the reviewer’s requirement, relevant statements have been added to in Section 7.

“Future research can be expanded from the following aspects: (1) increasing sensitivity analysis. In cost sensitivity analysis, adding uncertain demand to the expected number of vehicles, fuel costs under different load conditions of vehicles, and introducing the time-varying characteristics of road traffic in traffic environment sensitivity to affect transportation risks and rescue efficiency. (2) Consider the risk characteristics of different medical waste and the impact of climate conditions in different geographical regions on risk transmission. (3) In terms of algorithms, research on parallel processing of crossover and mutation operators can be strengthened to reduce the time complexity of the algorithm and improve its efficiency. At the same time, emerging technologies such as machine learning and artificial intelligence can be integrated, such as combining big data technology to analyze the impact of time-varying weather conditions and road traffic conditions on the risk of medical waste transportation, and combining deep reinforcement learning technology to optimize vehicle routes.”

5. In the NSGA-II algorithm, how is the initial population generated? What criteria are used to ensure diversity in the initial solutions?

According to the reviewer’s requirement, relevant statements have been added to in Section 5.3.

“(1) Initialize the population

The quality of initial feasible solutions will directly affect the efficiency of swarm intelligence optimization algorithms. In NSGA-II, a random generation method is used to generate the initial population, which may lead to insufficient population diversity and low convergence accuracy of the algorithm. To overcome this deficiency, this paper adopts the following improvement method to generate the initial population:

Step 1: Randomly generate a temporary population with a size of , use three objective functions as indicators to calculate the spatial distance between each chromosome and other chromosomes, and sort accordingly.

31

Step 2: Select chromosomes from in descending order to form the initial population.

(4) Crowding Distance

In NSGA-II, crowding distance is an important criterion for evaluating individual strengths and weaknesses. Crowding distance=(individual fitness - average fitness)/standard deviation. Under the same level of non dominated sorting, calculate the sum of the relative distances between a chromosome and its adjacent points in space at multiple objective function values. Crowding distance can ensure that the algorithm converges to a relatively uniform Pareto domain, achieving the goal of maintaining population diversity.”

6. W Ensure that all references are up-to-date and formatted according to the journal's guidelines. Some citations appear to be missing or incomplete. In general, the authors are recommended to refer to more recent papers in the manuscript, such as: doi.org/10.1016/j.eswa.2023.121035, doi.org/10.1109/TITS.2023.3285430.

According to the reviewer’s requirement, relevant statements have been added to in Section 2.

“Wang Nengmin et al. [20] defined the infection process and the transmission risk of infected populations, set up multiple disposal centers, developed a dual objective route model for medical waste collection to optimize infection risk and transportation costs, and obtained Pareto solution by combining weighted economic constraint method.”

7. The case study in Yi-bin City is mentioned, but details on the data used (e.g., waste generation rates, vehicle capacities) are sparse.

According to the reviewer’s requirement, relevant statements have been added to in Section 6.1.

“In addition, transportation risks have many uncertainties. To simplify the model in this paper, the following assumptions are made: the annual average wind speed in Yibin City is 4.43km/h. Probability of waste being infectious is 0.6; The population density of all lines in the region is 7500 person/km2; The vehicle height is 1 meter; The rescue capacity of the rescue center is 1000kg/h. Driving speed , vehicle load capacity in first echelon is 5t, fixed cost is 500 yuan, unit mileage transportation cost of 10 yuan, vehicle load capacity in second echelon is 1t, fixed cost is 200 yuan, unit mileage transportation cost is 5 yuan. ”

8. What assumptions did you make in your model regarding vehicle capacities, waste generation rates, and transit point locations? How might these assumptions impact the results?

According to the reviewer’s requirement, relevant statements have been added to in Section 4.2.

“(1) There is one medical waste disposal center, multiple transfer points, and multiple producing points in the area, and the disposal center is equipped with different types of vehicles to meet the transportation needs;

(2) Each producing point can only be served by one transport vehicle, and the amount of medical waste generated is less than the load capacity of the transport vehicle;”

9. Some figures and tables lack clear explanations in the text. Can you ensure that all visual aids are adequately referenced and explained?

According to the reviewer’s requirement, relevant statements have been added to in Section 6.2.

“By using Matlab to solve the problem, the average distance of each Pareto optimal solution tends to stabilize after about 160 iterations, and the Pareto front is shown in Figure 6. According to the selection strategy of Pareto solution, The recycling path of medical waste is shown in Figure 7. Among them, the total mileage of the vehicle is 92.62+18.99km, with a total cost of 1063+690 yuan and transport risk of 73.12 t·person/km3, requiring 3+1 vehicles (data before and after “+” are for the first and second echelons respectively).”

According to the reviewer’s requirement, relevant statements have been added to in Section 6.3.

“By using Matlab to solve the problem, the average distance of each Pareto optimal solution tends to stabilize after about 170 iterations, and the Pareto front is shown in Figure 8. The medical waste transport route of Pareto solution is shown in Figure 9. At this time, the total operating cost is 3249+1809 yuan, the transportation risk is 517.26 tons·person/km3, and the total mileage of vehicles is 89.78+30.87km, requiring 14+3 vehicles. The data before and after "+" represent the first and second stages, respectively.”

10. What are the limitations of your study, and how might they affect the applicability of your model in different contexts or regions?

According to the reviewer’s requirement, relevant statements have been added to in Section 7.

“However, this paper also has limitations, such as the lack of specialized qualifications and facility equipment requirements for the treatment of infectious medical waste in the study, and the possibility of irregular transportation strategies for transfer points that generate less medical waste. At the same time, this article did not consider the chemical and risk characteristics of different types of medical waste, and urban medical waste needs to be classified and treated according to different characteristics.”

11. What are the next steps for this research? Are there plans to extend the model to include other types of medical waste or different geographical areas?

According to the reviewer’s requirement, relevant statements have been added to in Section 7.

“Future research can be expanded from the following aspects: (1) increasing sensitivity analysis. In cost sensitivity analysis, adding uncertain demand to the expected number of vehicles, fuel costs under different load conditions of vehicles, and introducing the time-varying characteristics of road traffic in traffic environment sensitivity to affect transportation risks and rescue efficiency. (2) Consider the risk characteristics of different medical waste and the impact of climate conditions in different geographical regions on risk transmission. (3) In terms of algorithms, research on parallel processing of crossover and mutation operators can be strengthened to reduce the time complexity of the algorithm and improve its efficiency. At the same time, emerging technologies such as machine learning and artificial intelligence can be integrated, such as combining big data technology to analyze the impact of time-varying weather conditions and road traffic conditions on the risk of medical waste transportation, and combining deep reinforcement learning technology to optimize vehicle routes.”

Reviewer 2 Comments to the Author

Comments to Author:

1. I recommend the author to change the description of objective f_3 from section 4.4 to 4.2.3. It very hard and find where is the description of objective 3. Because the 4.1.1 and 4.2. is describe about f_1 and f_2.

According to the reviewer’s requirement, we change the description of objective f_3 from section 4.4 to 4.3.3

4.3.3 Rescue model in case of infectious medical waste leakage

When medical waste belongs to highly contagious, widely contagious, and rapidly contagious infectious substances, m is the location of infectious waste leakage, . At this point, add a new objective function : the fastest rescue speed, i.e. the shortest rescue distance:

12

2. This paper use NGSA-II to find the 3 objectives optimization problem. However, I cannot see you Pareto solutions of your problem. I recommend the author to added the figure of Pareto solutions in your manuscript.

According to the reviewer’s requirement, relevant statements have been added to in Section 6.2.

By using Matlab to solve the problem, the average distance of each Pareto optimal solution tends to stabilize after about 160 iterations, and the Pareto front is shown in Figure 6. According to the selection strategy of Pareto solution, The recycling path of medical waste is shown in Figure 7. Among them, the total mileage of the vehicle is 92.62+18.99km, with a total cost of 1063+690 yuan and t

---

## [Decision Letter · Decision Letter 1]

14 May 2025

Dear Dr. Chen,

Thank you for submitting your manuscript to PLOS ONE. After careful consideration, we feel that it has merit but does not fully meet PLOS ONE’s publication criteria as it currently stands. Therefore, we invite you to submit a revised version of the manuscript that addresses the points raised during the review process.

We look forward to receiving your revised manuscript.

Kind regards,

Erfan Babaee Tirkolaee, PhD

Academic Editor

PLOS ONE

Reviewers' comments:

Reviewer's Responses to Questions

**Comments to the Author**

Reviewer #2: All comments have been addressed

Reviewer #3: All comments have been addressed

Reviewer #4: (No Response)

2. Is the manuscript technically sound, and do the data support the conclusions?

Reviewer #2: Yes

Reviewer #3: Yes

Reviewer #4: Yes

3. Has the statistical analysis been performed appropriately and rigorously?

Reviewer #2: Yes

Reviewer #3: I Don't Know

Reviewer #4: Yes

4. Have the authors made all data underlying the findings in their manuscript fully available?

Reviewer #2: Yes

Reviewer #3: Yes

Reviewer #4: No

5. Is the manuscript presented in an intelligible fashion and written in standard English?

Reviewer #2: Yes

Reviewer #3: Yes

Reviewer #4: Yes

Reviewer #2: I agree with the author's response, and all of comments have been addressed. Now the paper is ready for publication.

Reviewer #3: Manuscript ID: PONE-D-3149R1, entitled "Research Optimization of Transportation Routes for Infectious Medical Waste".

The manuscript presents a timely and pertinent contribution to the field by proposing a multi-objective optimization model for the routing of infectious medical waste. While the topic is significant and the methodological approach is well-motivated, there are several aspects of the manuscript that require further refinement to meet the standards of academic publication.

1. Abstract

The abstract would benefit from a clearer emphasis on the main numerical findings of the study. Explicitly stating quantitative results will strengthen the perceived value of the work.

2. Introduction

The introduction should avoid vague qualifiers such as "important issue" and instead provide quantified evidence, preferably supported by current references or statistics. Additionally, the articulation of research gaps needs to be more prominent. The authors are encouraged to use visual or structural enhancements—such as bullet points or bolded text—to emphasize the novelty and motivation of their work.

3. Literature Review

The literature review section identifies several relevant studies, but does so without critical analysis. It is recommended that the authors include a summary table comparing key features of the most closely related works with their own proposed approach. This will help clarify the distinct contributions of the current study.

4. Solution Methodology

In the methodology section, it is important to justify the choice of the improved NSGA-II over other leading multi-objective evolutionary algorithms (MOEAs). A comparative rationale, perhaps supported by relevant citations or empirical evidence, would enhance the credibility of the algorithmic selection.

5. Numerical Experiments

Figures should be more clearly referenced and interpreted within the main text. Furthermore, the use of units and performance metrics should be standardized and clearly defined throughout the experimental analysis.

6. Conclusion

The limitations section is appreciated and a good starting point. However, it would benefit from further elaboration to include:

• Real-time traffic uncertainty;

• Dynamic generation rates of medical waste;

• Regulatory constraints and compliance issues related to infectious waste logistics.

Reviewer #4: In the introduction, you need to connect the state of the art to your paper goals. Please follow the literature review by a clear and concise state of the art analysis. This should clearly show the knowledge gaps identified and link them to your paper goals. Please reason both the novelty and the relevance of your paper goals. Clearly discuss what the previous studies that you are referring to. What are the Research Gaps/Contributions? Please note that the paper may not be considered further without a clear research gap and novelty of the study and compare your paper with municipal cooperative waste supply chain papers and sustainable medical waste supply chain network

Constraints such as “uniqueness,” “logicality,” and “load capacity” are vaguely defined. Can you explicitly formulate these constraints mathematically within the model and explain their operational relevance?

The paper states it adopts a "segmented strategy" for modeling without adequately explaining what this segmentation entails or how it improves over a holistic model. Could you elaborate on what stages are modeled and how segmentation affects the computational efficiency or route design?

While the improved NSGA-II is claimed to outperform a traditional genetic algorithm, no formal statistical validation or benchmark comparison is shown. Could the authors compare with at least one other modern multi-objective algorithm (e.g., MOEA/D, SPEA2) and report metrics such as hypervolume or spread?

The problem instances and solution parameters (e.g., 58 production points, locations, transfer point limits) are not fully detailed. Can the authors provide a table summarizing the data and parameters used in simulations to ensure replicability?

The transport risk is quantified as “t·person/km³,” which is unclear and lacks explanation. What is the theoretical or empirical basis for this risk metric, and how does it influence decision-making in route optimization?

The methodology and results sections contain excessive narrative text and insufficient formalism (e.g., pseudocode, mathematical notation). Would the authors consider rewriting key parts using clear pseudocode for the algorithm and structured tables for solution outcomes?

**Do you want your identity to be public for this peer review?** For information about this choice, including consent withdrawal, please see our Privacy Policy

Reviewer #2: No

Reviewer #3: No

Reviewer #4: No

---

## [Author Response · Author response to Decision Letter 2]

25 Jul 2025

Dear reviewer and editor,

Thank you for allowing a resubmission of our manuscript, with an opportunity to address the reviewers’ comments.

We are uploading (a) our point-by-point response to the comments (below) (response to reviewers), (b) an updated manuscript with yellow highlighting indicating changes, and (c) a clean updated manuscript without highlights (Word main document).

Best regards

---

## [Decision Letter · Decision Letter 2]

10 Aug 2025

Research on Optimization of Transportation Routes for Infectious Medical Waste

PONE-D-24-35149R2

Dear Dr. Chen,

We’re pleased to inform you that your manuscript has been judged scientifically suitable for publication and will be formally accepted for publication once it meets all outstanding technical requirements.

Kind regards,

Erfan Babaee Tirkolaee, PhD

Academic Editor

PLOS ONE

Additional Editor Comments (optional):

Reviewers' comments:

Reviewer's Responses to Questions

**Comments to the Author**

Reviewer #3: All comments have been addressed

Reviewer #4: All comments have been addressed

2. Is the manuscript technically sound, and do the data support the conclusions?

Reviewer #3: Yes

Reviewer #4: Yes

3. Has the statistical analysis been performed appropriately and rigorously?

Reviewer #3: Yes

Reviewer #4: Yes

4. Have the authors made all data underlying the findings in their manuscript fully available?

Reviewer #3: Yes

Reviewer #4: Yes

5. Is the manuscript presented in an intelligible fashion and written in standard English?

Reviewer #3: Yes

Reviewer #4: Yes

Reviewer #3: (No Response)

Reviewer #4: No comments

No comments

No comments

No comments

No comments

No comments

No comments

No comments

No comments

**Do you want your identity to be public for this peer review?** For information about this choice, including consent withdrawal, please see our Privacy Policy

Reviewer #3: No

Reviewer #4: No

---

## [Editor Report · Acceptance letter]

PONE-D-24-35149R2

PLOS ONE

Dear Dr. Chen,

I'm pleased to inform you that your manuscript has been deemed suitable for publication in PLOS ONE. Congratulations! Your manuscript is now being handed over to our production team.

Kind regards,

on behalf of

Dr. Erfan Babaee Tirkolaee

Academic Editor

PLOS ONE